

# Towards a typology for hybrid compound flood modeling

Soheil Radfar[1,2*], Hamed Moftakhari[1,2*], David F. Muñoz[3], Avantika Gori[4], Ferdinand Diermanse[5], Ning Lin[6], Amir AghaKouchak[7]

[1]Center for Complex Hydrosystems Research, The University of Alabama, Tuscaloosa, AL, USA.

[2]Department of Civil, Construction and Environmental Engineering, The University of Alabama, Tuscaloosa, AL, USA.

[3]Department of Civil and Environmental Engineering, Virginia Tech, Blacksburg, VA, USA

[4]Department of Civil and Environmental Engineering, Rice University, Houston, TX, USA

[5]Deltares, Delft, the Netherlands

[6]Department of Civil and Environmental Engineering, Princeton University, Princeton, NJ, USA

[7]Department of Civil and Environmental Engineering, University of California, Irvine, CA, USA

*Correspondence to*: Soheil Radfar (sradfar@ua.edu) and Hamed Moftakhari (hmoftakhari@eng.ua.edu)

**Abstract.** Modeling compound flood events requires sophisticated approaches that can capture complex nonlinear interactions between multiple flood drivers. While combining different data-driven and physics-based modeling approaches has shown promise, the criteria for classifying such combinations and the underlying terminology to describe them remain
inconsistent in the literature. To establish classification criteria, we introduce a systematic framework for defining and categorizing hybrid physical-statistical modeling approaches in compound flood modeling. Hybrid compound flood models offer significant advantages in terms of prediction accuracy and computational efficiency over traditional single-model approaches, particularly in coastal regions where multiple flooding mechanisms frequently interact. Here, we introduce a systematic framework for defining hybrid models and establish clear classification criteria based on their structural and
functional characteristics. We identify three categories of hybrid models: sequential, feedback, and ensemble. Through illustrative examples, we demonstrate how each category leverages the strengths of its component models while also maintaining their independence. The proposed framework enables a systematic evaluation of different hybrid modeling strategies, enhancing model comparability and supporting the development of more effective compound flood prediction tools.

## 1 Introduction

Flood modeling is undergoing a paradigm shift as researchers and practitioners recognize the limitations of traditional approaches that analyze individual flooding mechanisms in isolation. Historically, these approaches relied on univariate





statistics, which are straightforward but inherently simplistic. For instance, in the United States, riverine flood inundation mapping typically employs peak volumetric flow rates at specific return periods obtained from the log-Pearson type III

distribution. Those flow rates with certain return periods then  inform the upstream boundary conditions for hydraulic and/or hydrodynamic models to generate flood inundation maps (England Jr et al., 2018). Similarly, coastal flooding assessments rely on the probability distribution of extreme water levels at the coast, calculated based on the Extreme Value theorem, to delineate flood hazard zones (Fema, 2018, 2016). While effective for single-hazard scenarios, univariate methods fall short in addressing the complexities of compound flooding in low-lying coastal areas, where multiple flooding mechanisms, such

as coastal, pluvial, and fluvial processes, nonlinearly interact in flood transition zones (Bilskie and Hagen, 2018; Han and Tahvildari, 2024). Compound events, such as multi-mechanism floods, can produce more severe impacts than individual mechanisms in isolation (Aghakouchak et al., 2020; Zscheischler et al., 2018; Bevacqua et al., 2021). Numerous examples highlight the shortcomings of traditional univariate approaches. For instance, Hurricane Harvey in 2017 caused catastrophic flooding in the Houston metropolitan area, where simultaneous pluvial and fluvial flooding overwhelmed infrastructure

designed based on univariate flood estimates (Harr et al., 2022; Lan et al., 2022). Similarly, during Hurricane Sandy in 2012, interactions between storm surge and riverine flooding exacerbated flood damage in New York and New Jersey, demonstrating the inadequacy of single-hazard approaches for coastal megacities (Goulart et al., 2024). As such, univariate methods are inadequate for capturing interactions between multiple flooding drivers and do not consider the statistical dependence (i.e., joint probability) between multiple flood drivers. To fully capture the statistical dependence of flood

drivers and resolve their dynamics in coastal zones, it is necessary to integrate multivariate statistical methods with physics-based models. The integration of statistical and physics-based methods can overcome data limitations and improve flood risk assessments (Cho et al., 2023; Nasr et al., 2021).

The transition from univariate hazard assessment to a probabilistic multivariate risk assessment introduces its own set of challenges. Multivariate parametric distributions (i.e., Gaussian, t, gamma, extreme value) have traditionally been used to

extend univariate models to higher dimensions, but they are constrained by limited options for marginal distribution selection and parameterization, and struggle to capture nonlinear dependence structures (Hao and Singh, 2016). Copulas address these shortcomings by providing a more flexible framework for representing interdependencies among multiple flood drivers (Salvadori and De Michele, 2004, 2007; Durante and Sempi, 2016). The fundamental challenge, however, lies in obtaining sufficient length (usually decades) of overlapping (both in time and space) records of multiple compound flood

drivers to detect and reflect their interdependencies. Such information is often unavailable, particularly in regions where observational infrastructure is sparse (Ward et al., 2018). Even in well-monitored regions, most gauge networks lack long continuous records to capture the variability of interdependence dynamics between coastal, fluvial, and pluvial processes, resulting in fragmented datasets that cannot adequately characterize joint probabilities. While long historical records may exist, data-driven methods face major limitations due to their underlying assumptions, like stationarity, which restricts their

ability to account for evolving anthropogenic influences such as climate change (Radfar et al., 2023) and land use modifications (Long and Duan, 2025) that can alter the probability of compound flooding events producing impacts greater



than individual mechanisms in isolation. Furthermore, statistical methods usually characterize dependence at point locations, raising the question of how to translate point estimates of dependence to full 2-D floodplains for comprehensive flood risk assessment.

In such cases, researchers rely on physics-based numerical models, which use gauge measurements or climatological hydrodynamic stimulations as boundary conditions to simulate spatially distributed flood characteristics. While these models are invaluable for understanding the dynamics of compound flooding and informing resilience planning, they have inherent limitations. For example, significant assumptions in model formulation and parameterization can introduce uncertainties, potentially undermining the accuracy of predictions (Muñoz et al., 2022a; Jafarzadegan et al., 2021). Moreover, applying

these models to large domains at high spatial resolutions for flood delineation is computationally expensive, often requiring high-performance computing systems and resources to analyze multiple scenarios. Additionally, to capture the complex interplay between natural processes underlying compound floods, model coupling (i.e., integrating hydrologic and hydrodynamic systems) is frequently necessary. This coupling adds another layer of complexity to the modeling process. In fact, coupled flood modeling presents significant challenges due to the intricate interdependence of hydrologic and

hydrodynamic processes (Santiago-Collazo et al., 2019). Hydrologic models are employed to estimate the timing and magnitude of runoff generated by rainstorms over a watershed. These models rely on routing schemes to convey runoff to the watershed outlet, producing an outflux that serves as input to hydrodynamic models established on downstream floodplains. However, a critical complexity arises from the frictional slope in the routing scheme, which is a key determinant of the water volume that can flow through open channels. The frictional slope is influenced by downstream boundary

conditions, such as water levels, which are unknown prior to running hydrodynamic models. Conversely, the hydrodynamic model requires the routed flux from the hydrologic model as an upstream boundary condition. This two-way dependency underscores a major hurdle: identifying downstream water levels before executing the hydrodynamic model, while also needing upstream flux estimates from the hydrologic model. Addressing this challenge requires innovative approaches to improve model integration and streamline the feedback loop between hydrologic and hydrodynamic components.

Efforts to address these challenges have highlighted the potential of integrating statistical and physics-based modeling approaches. Such methodology capitalizes on the complementary strengths of each approach: the statistical models' ability to handle multivariate dependencies and the physics-based models' capacity to simulate physical processes in detail. By connecting these methodologies, researchers can produce more reliable compound flood risk assessments while maintaining computational efficiency (Moftakhari et al., 2019; Muñoz et al., 2020; Serafin et al., 2019; Gori and Lin, 2022). In recent

years, there has been growing recognition that combined approaches, in what might be called "linked," "coupled," or "hybrid" frameworks, can improve the accuracy, efficiency, robustness, and interpretability of model outputs. In the compound flood research community, the term "linked" models usually refers to one-way transfer of information between physics-based models in which outputs from one serve as boundary conditions for another, whereas "coupled" models mainly refer to two-way or fully integrated frameworks (loose, tight, or fully coupled) where information flows in both

directions or the codes are merged (Santiago-Collazo et al., 2019; Xu et al., 2023). By contrast, in hydrology and earth



system science, any integration of physics-based and statistical/data-driven methods is often considered a "hybrid model" (Kraft et al., 2021; Reichstein et al., 2019). This broader usage highlights the need for a more precise definition tailored to compound flood applications.

The criteria for which we can call a model setup "hybrid" and the different approaches to develop such model frameworks remain unclear in the community. This lack of clear classification criteria has led to inconsistent terminology in the literature, where terms like "hybrid", "coupled," and "linked" are often used interchangeably. Such ambiguity makes it challenging to compare different modeling approaches, evaluate their relative strengths and limitations, and identify best practices for specific applications. Furthermore, without standardized definitions, it becomes difficult to systematically assess the added value of different hybrid modeling strategies in addressing the compound flood challenges. In this paper, we

introduce a conceptual framework and clear definitions for how we can categorize combinations of physics-based and statistical models, focusing on the context of compound flooding. The structured classification of hybrid models presented aims to provide a unified typology and foundation for evaluating and applying hybrid methodologies in compound flood modeling.

## 2 Defining hybrid models

The modeling community has developed various approaches to simulate compound flooding, ranging from physics-based methods that capture physical phenomena to statistical and machine learning techniques that leverage historical data patterns. When these approaches are combined to address the complex nature of compound flooding (Figure 1), they create what we broadly term "hybrid models." The defining characteristic of hybrid models is the integration of at least one physics-based component with at least one statistical or data-driven component, where each maintains its specialized methodology (i.e., is

process-specific) while contributing to a more complete representation of compound processes. In this definition, "process-specific" means the component is designed to represent one or more flood drivers or subprocesses (e.g., atmospheric forcing, watershed runoff and routing, river discharge, coastal surge, or wave dynamics). For instance, in coastal flooding applications, a statistical model might generate probabilistic storm scenarios or boundary conditions, while a physics-based hydrodynamic model simulates the detailed flood dynamics. When combined, these components form a hybrid system

because they unite different modeling paradigms, leveraging both the mechanistic understanding of physics-based approaches and the probabilistic capabilities of statistical methods. It is noteworthy that we treat a component model as part of a hybrid modeling framework only if it (*i*) acts before or during the execution of the other component(s) (i.e., supplies boundary forcing or performs runtime state updates to the other model) and (*ii*) injects new variability, probabilistic structure, or state corrections that the other component(s) would not otherwise produce (e.g., synthetic joint forcings,

probabilistic ensembles, or assimilated state corrections). Therefore, utilities that simply reshape, rescale, or re-grid an existing series without injecting new variability or cross-variable structure (e.g., spatiotemporal interpolation, bias/trend



adjustments, or other similar post-run tweaks) are treated as pre-/post-processors, not hybrid components. For example, a machine learning model can be considered a valid hybrid component if it generates reliable predictions, such as water level forecasts or rainfall-runoff patterns, without requiring continuous guidance from external models.

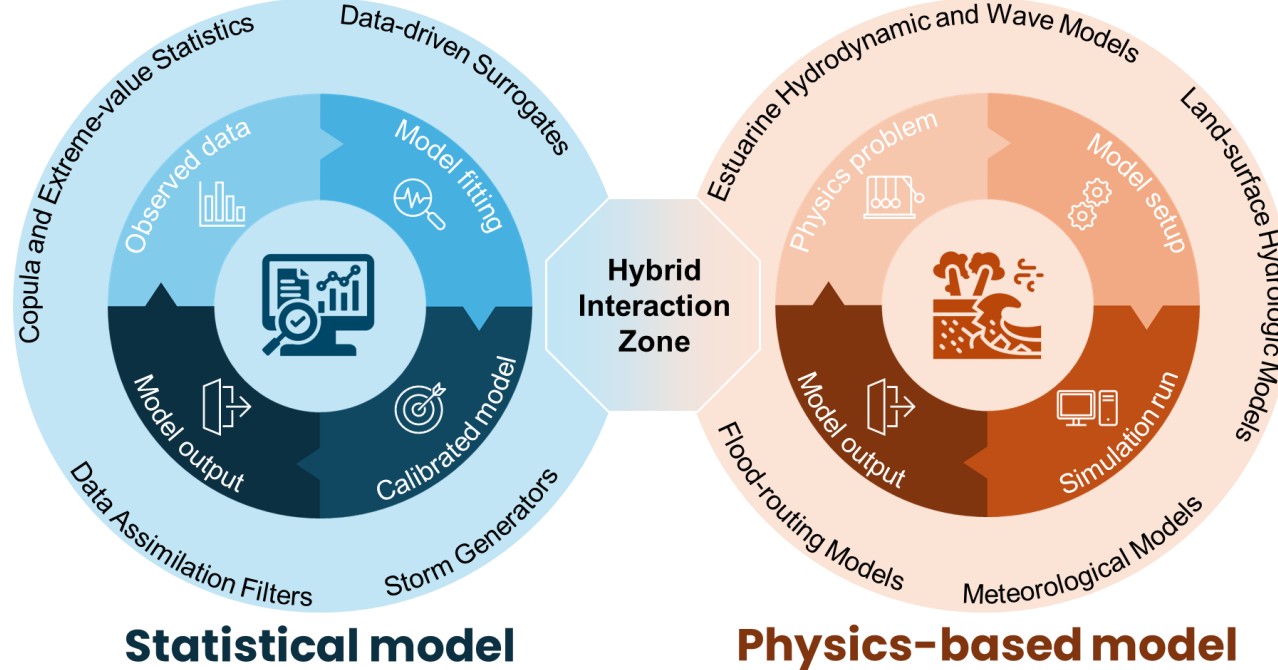

**Figure 1. Conceptual representation of hybrid physical-statistical models for compound flooding.** Statistical/data-driven models (left) include methods such as copulas, storm generators, and ML surrogates that transform observed data into calibrated models and outputs. Physics-based models (right) include hydrodynamic, hydrologic, wave, routing, and meteorological models that solve physical problems through model setup and simulation. The hybrid interaction zone (center) highlights where the two paradigms exchange information.

This separation of process-specific components makes the overall framework more robust and flexible. Because each model has been validated for the drivers or subprocesses it represents, the integrated system inherits those proven capabilities without forcing one component to compensate for weaknesses in another. The result is a more resilient approach to modeling compound flooding, where the strength of the whole system builds upon the proven reliability of its parts. Hybrid models vary significantly in both their complexity and how information flows between components. At the simpler end of the spectrum, one model may provide data directly to another (e.g., when regional climate projections are translated into local flood scenarios). More sophisticated implementations involve intricate processing chains, where data undergoes multiple transformations before and after each modeling stage. Information flow can therefore be as straightforward as an upstream-to-downstream hand-off or as intricate as a two-way exchange that repeatedly refines shared state variables.

Regardless of complexity, the hybrid approach enables practitioners to combine complementary strengths—physical realism from the physics solver and probabilistic or surrogate efficiency from the statistical block—into a single, compound



flood modeling system. The true value of hybrid modeling emerges from its ability to combine the distinct advantages of different approaches while preserving what makes each component effective. Consider how a physics-based model can capture the intricate dynamics of water flow, while a statistical approach might efficiently quantify hazard scenarios, uncertainties, and generate fast predictions. By bringing these complementary capabilities together in a hybrid framework,

we can achieve more comprehensive simulations of compound flooding than would be possible with single-model approaches.

## 3 Types of hybrid models

The growing complexity of compound flooding challenges has led to diverse implementations of hybrid models, each tailored to specific needs and constraints. These implementations vary not only in how they handle information flow between

components but also in their computational demands and level of integration. To systematically understand these approaches, we classify hybrid models into three fundamental categories: sequential, feedback, and ensemble (Figure 2). This classification is based on how information flows between components and how their individual capabilities are combined to create a comprehensive modeling system.

These categories represent distinct paradigms for model integration, each with its own advantages and challenges. The

choice between these approaches often depends on factors such as the physical processes being modeled, computational resources available, and the specific requirements of the application. Each approach represents a different strategy for combining cross-paradigm models while maintaining their individual integrity and capabilities. Throughout this paper, "hybrid" refers specifically to frameworks that combine process-specific statistical (data-driven) components with physics-based models. The upcoming sections explore different types of hybrid models with conceptual examples and discuss their

relative strengths and applications in compound flood modeling. Appendices A to C include illustrative examples from recent literature for each model. Although we introduce this typology for compound flooding, the same sequential, feedback, and ensemble logic can be applied to other compound hazards across hydrology, atmospheric science, and the broader Earth and environmental science disciplines.





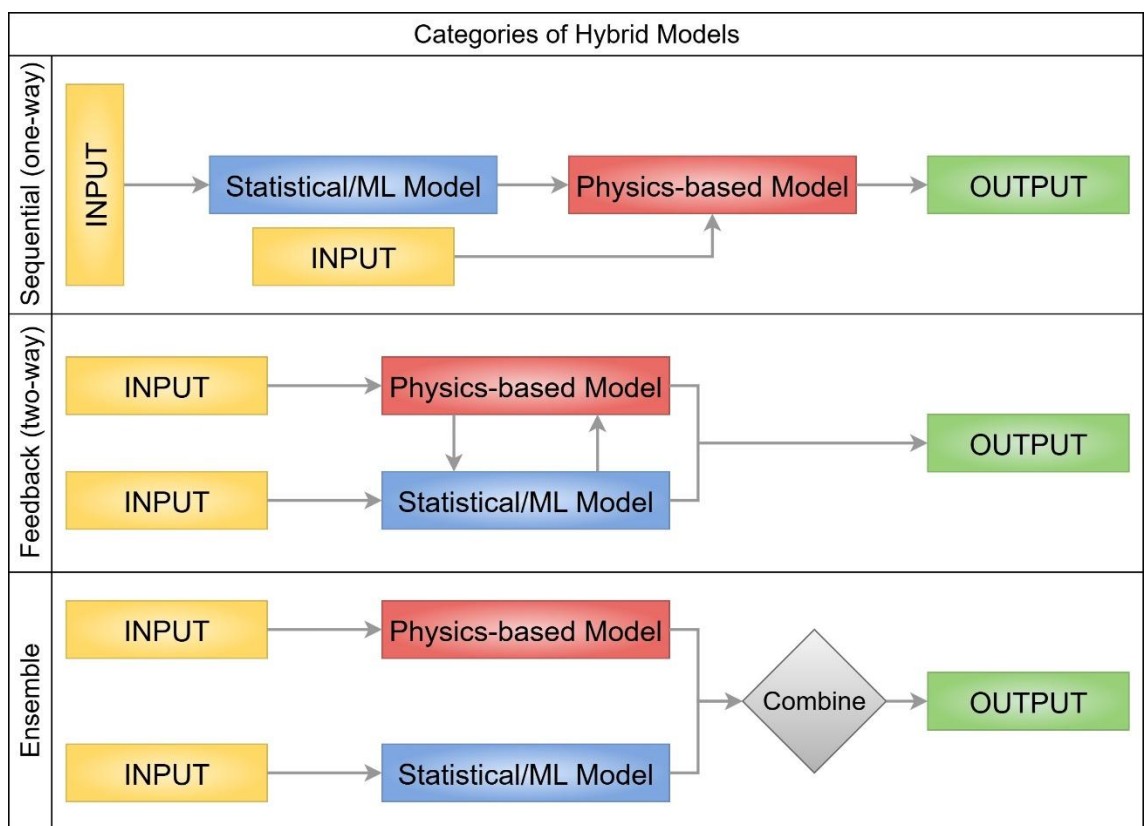

**Figure 2. Schematic illustration of three categories of hybrid models: sequential (one-way), feedback (two-way), and ensemble approaches.** Each category integrates statistical/data-driven and physics-based models differently. In sequential models, information flows unidirectionally from one model to the other. In feedback models, the two components exchange information bidirectionally throughout the simulation. In ensemble models, both components run independently in parallel, and their outputs are combined through an aggregation algorithm. Gray arrows indicate information flow between components.

## 3.1 Sequential Models

A sequential hybrid model is characterized by information flowing predominantly in one direction, from one process-specific model to another, in a linear chain. Despite its one-way structure, each model remains comprehensive within its domain. An example might involve a data-driven rainfall-runoff model that generates discharge predictions from meteorological data. This model can simulate the rainfall-runoff transformation without external assistance. The generated discharge values then serve as inputs to a hydraulic model that can independently simulate flood routing and inundation. Because the chain includes cross-paradigm integration of two process-specific models, the system qualifies as a hybrid model. A similar case arises when a validated statistical surge model (capable of producing nearshore surge estimates) is sequentially coupled to a





coastal flooding model that needs storm surge levels as boundary input to delineate the flood zone under a given set of forcing.

Sequential hybrid models can be illustrated through compound flood modeling studies where river discharge and storm surge interactions are simulated (Figure 3). In a typical setup, a riverine model simulates river dynamics using various inputs, including upstream flow measurements, infiltration capacity, and local rainfall data. The output, consisting of river discharge, is then passed as input to a hydrodynamic model developed over the transition zone, which combines the riverine input with wind fields, atmospheric pressure, and tidal conditions to simulate compound flooding effects. This one-way

transfer of information from the riverine model to the coastal model, where each model maintains its full functionality in its respective domain while information flows predominantly in one direction, exemplifies a sequential hybrid modeling framework. Figure A1 presents a detailed example of sequential hybrid models where multiple statistical and hydrologic-hydrodynamic models are one-way coupled together.

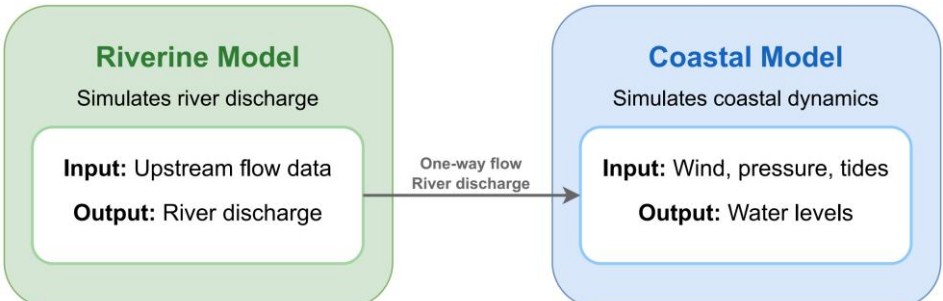

**Figure 3. Schematic illustration of a sequential hybrid modeling framework for simulating compound flooding in coastal river systems.** The riverine model (left) acts as a complete simulator processing upstream flow data to generate river discharge. The coastal model (right) serves as a complete simulator combining multiple inputs, including wind fields, atmospheric pressure, and tidal conditions, with the river discharge inputs. Gray arrow indicates one-way information flow from the riverine to coastal model, characteristic of sequential hybrid approaches where both components maintain complete simulation capabilities in their respective domains while

information flows predominantly in one direction.

The one-way transfer of information has paved the way for the widespread application of sequential hybrid physical–statistical models in compound flood research. They are conceptually straightforward and do not require complex two-way data exchange, allowing for efficient simulation of compound flooding events while maintaining the specialized capabilities of each model component. In these frameworks, the physics engine remains intact while a separate statistical layer prescribes

boundary conditions or supplies a catalogue of extreme events. This approach preserves the mechanistic integrity of physics-based models while leveraging statistical methods to overcome data limitations and extend scenario coverage beyond historical observations. They are particularly useful in situations where the influence of one process on another is predominantly unidirectional, such as when river discharge affects coastal dynamics, and tidal influences (e.g., backpropagation) do not significantly impact upstream river conditions. Similarly, they are suitable when meteorological





conditions (rainfall and wind) impact river discharge and surge, but the flood response does not impact the meteorological conditions.

Although the statistical block in a sequential hybrid can also be a parametric tropical-cyclone (TC) rainfall generator that converts synthetic tracks into space–time precipitation fields (Gori and Lin, 2022; Gori et al., 2020; Gori et al., 2022; Bilskie et al., 2021), multivariate statistical frameworks have formed the foundation for many sequential hybrid

implementations, with copula-based approaches enabling the extraction of joint extremes from historical surge, rainfall, and river discharge records to drive physics-based models through comprehensive event catalogues (Xu et al., 2024; Gao et al., 2023; Olbert et al., 2023). These methodologies have demonstrated versatility across diverse coastal environments and applications, from urban coastal settings to estuarine flood mapping, wetland impact assessment, and navigation studies by using methods like trivariate copula feeding two-dimensional inundation solvers (Wang et al., 2025), copula-based Bayesian

networks and Gaussian-process emulators for translating joint extremes into flood depths across multiple catchments (Couasnon et al., 2018; Bass and Bedient, 2018), and bivariate copula samplers providing surge-river pairs to force Delft3D-FM for wetland-elevation effects and vessel under-keel clearance studies along the Gulf Coast (Muñoz et al., 2020; Muñoz et al., 2022b).

Recent developments have expanded sequential hybrid approaches by integrating them with large-scale storm

climatology to generate probabilistic surge and hydrological conditions to support comprehensive flood hazard assessments. These implementations demonstrate the scalability of sequential approaches through strategic combinations such as GTSR surge hindcast pairing with LISFLOOD peaks via copulas (Eilander et al., 2023), tropical cyclone rainfall model outputs, and ADCIRC surge simulations combined in a Bayesian joint probability framework (Gori and Lin, 2022; Gori et al., 2020; Gori et al., 2022), TELEMAC simulations linked to Gumbel copulas for Mediterranean applications (Zellou and Rahali,

2019), and ROMS surge hindcasts paired with rainfall data via stochastic resampling for estuarine-extreme quantification (Wu et al., 2018; Wu et al., 2021). The incorporation of stochastic event generation, ranging from parametric weather libraries that generate pseudo-global-warming typhoons for Delft3D (Toyoda et al., 2025) or perturb thousands of cyclone tracks through a coupled tide–surge–rainfall model to be used with SFINCS (Nederhoff et al., 2024), to Monte-Carlo resampling of historical surge-runoff pairs that drive a cross-sectional Delft3D solver along river mouths (Serafin et al.,

2019), to physics-based statistical modeling of climate-variant synthetic tropical cyclones (Jing and Lin, 2020), has further enhanced scenario diversity.

ML integration marks a major step forward in sequential hybrid modeling, where statistical surrogates replace computationally intensive physics-based components while keeping prediction accuracy. These implementations demonstrate computational efficiency gains through approaches such as Gaussian-process emulators replacing

WaveWatch3-SWAN-FLOW(Delft3D)-XBeach simulation chains (Anderson et al., 2021), treed Gaussian processes derived from SFINCS runs evaluating ten thousand synthetic events (Terlinden-Ruhl et al., 2025), CatBoost models trained on ADCIRC-HEC-RAS libraries for direct coastal-plain inundation prediction (Huang, 2022), random forest surrogates merging tidal and pluvial flooding (Zahura and Goodall, 2022), and deep learning models predicting the evolution of compound flood





dynamics (e.g., flood depth and extent over time) attributed to cyclones within coastal systems (Daramola et al., 2025).
Together, these studies demonstrate how sequential hybrids translate sophisticated statistics—whether copulas, storm
generators, or active-learning surrogates—into flood-ready boundary conditions or rapid hazard catalogues.

## 3.2 Feedback Models

Feedback-driven hybrid models allow two or more process-specific models to exchange information in an iterative or two-
way manner. This category captures scenarios in which one physical process directly affects another, prompting updates on
both sides. For example, in coastal systems, water level predictions from an ocean model can influence wave calculations,
while wave-induced forces can affect water circulation patterns. Similarly, in atmospheric-land surface interactions, changes
in soil moisture can influence local weather conditions, while atmospheric conditions determine precipitation and
temperature patterns that affect the land surface. Figure 4 illustrates a feedback hybrid model that integrates atmospheric,
land, and ocean processes. In such systems, multiple complete models, each capable of independently simulating its domain,
exchange data bi-directionally at specified time intervals. The atmospheric model provides essential forcing data, such as
wind stress, air pressure, and precipitation, to the ocean model, and precipitation, temperature, wind speed, radiation, and
pressure to the land surface and river models. The ocean model can also supply wave-state information back to the
atmospheric component. Land-and-river models route rainfall to generate runoff/baseflow and pass river discharge to the
ocean model, while the ocean model returns water levels at river mouths. This feedback mechanism allows each model to
respond to changes in other components, capturing nonlinear interactions critical for compound flooding.

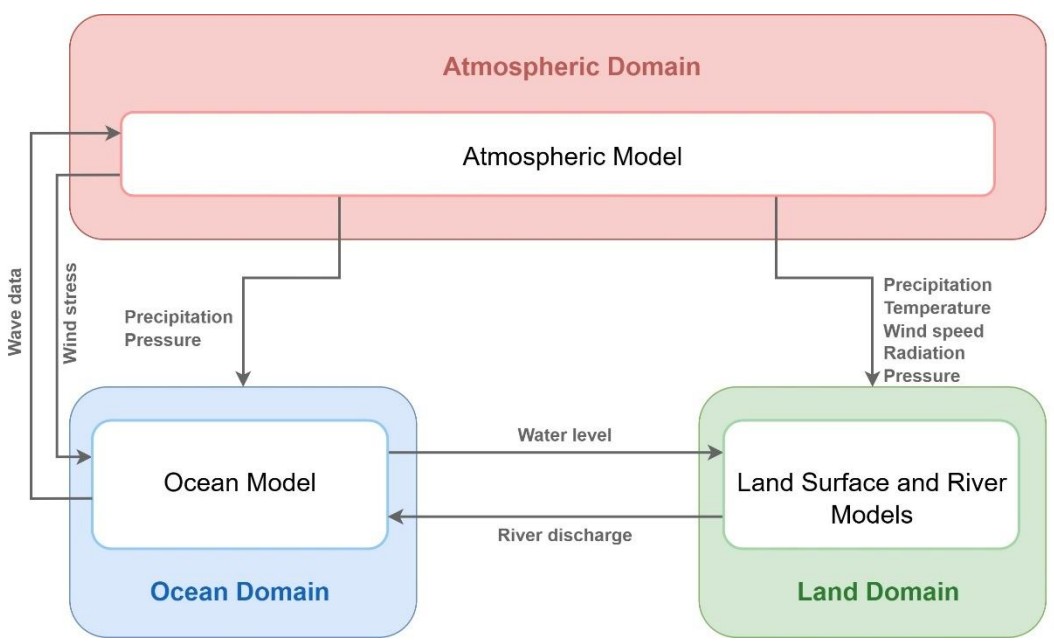



**Figure 4. Schematic illustration of a feedback hybrid modeling framework for simulating environmental processes and their interactions.** The framework shows three main domains (atmospheric, land surface, and ocean) with their respective models exchanging information bi-directionally. Colored boxes represent different environmental domains, while arrows indicate the flow and type of information exchanged between models (e.g., precipitation, river discharge, water levels). This feedback coupling enables the representation of complex interactions between atmospheric, land surface, and oceanic processes that are essential for understanding compound flooding dynamics. Each model maintains complete simulation capabilities in its domain while exchanging essential data with other models to capture the coupled nature of the system.

The key advantage of feedback hybrid models is their ability to capture complex interactions between different environmental processes. By allowing models to communicate and update their states based on information from other components, these frameworks can represent the dynamic nature of compound flooding more realistically than sequential approaches. This is particularly important in coastal regions where river discharge, tidal effects, and storm surge can interact in complex ways to influence flooding patterns.

While the bidirectional information exchange characteristic of feedback hybrid models is a relatively recent development in compound flooding research, the broader compound flood literature has extensively benefited from linking two-way or tightly coupled physics-based models (Radfar et al., 2024; Santiago-Collazo et al., 2019). Two-way (or loosely) coupled models involve models that communicate through iterative information exchange while running separately. In this context, tightly coupled models integrate the source codes of independent models, and information exchange happens at each computational time step (Green et al., 2025; Santiago-Collazo et al., 2019). Numerous studies have focused on developing tightly coupled wave and ocean circulation models, particularly ADCIRC and SWAN models (Tejaswi et al., 2025; Cassalho et al., 2023; Holzenthal et al., 2022; Bilskie et al., 2022; Rezaie et al., 2021; Zhu et al., 2020; Serafin et al., 2019; Bigalbal et al., 2018; Deb and Ferreira, 2017; Brandon et al., 2016; Lawler et al., 2016; Kerr et al., 2013). In these implementations, the wave model SWAN (Booij et al., 1999) supplies wave radiation stresses and spectral information to the circulation model ADCIRC (Luettich and Westerink, 2004), which returns updated water surface elevations and velocity fields to influence wave calculations. Delft3D-FM represents another comprehensive framework for physics-based coupling (Han and Tahvildari, 2024; Vona and Nardin, 2023; Velasquez-Montoya et al., 2022; Kupfer et al., 2022) that couples flow, waves, and rainfall-driven processes through its integrated D-Waves module, which incorporates the SWAN model to simulate wave field evolution, atmospheric wave generation, higher-order spectral interactions, and wave attenuation through bottom friction and vegetation effects (Deltares, 2022).

Beyond wave-circulation coupling, two-way physics-based coupling has been widely used for hydrodynamic-hydrologic interactions in compound flood scenarios. This approach recognizes the fundamental bidirectional nature of coastal-riverine systems, where surge-driven backwater effects influence upstream discharge while riverine inflows simultaneously modify coastal circulation patterns. The implementation of such coupling typically involves iterative exchange of boundary conditions at predefined interface points and time intervals, enabling dynamic representation of processes that span the terrestrial-marine transition zone. The evolution of hydrodynamic-hydrologic coupling has progressed from regional



implementations connecting ocean circulation and watershed models for hurricane impact assessment (Cheng et al., 2010) to sophisticated global-scale frameworks that integrate river routing networks within Earth-system models (Feng et al., 2024). These systems enable dynamic exchange of mass and momentum between terrestrial and marine domains through mechanisms in which land-surface runoff feeds ocean models that then return surge-induced elevation changes to update riverine stages. The complexity of such implementations has evolved to include comprehensive multi-domain frameworks that integrate atmospheric, terrestrial, and oceanic physics-based models through coordinated coupling strategies with varying temporal exchange frequencies, such as (Zhang and Yu, 2025)'s five-model system that integrates ADCIRC and SWAN with dynamic coupling, CaMa-Flood for river routing with hourly ocean boundary interactions, Variable Infiltration Capacity (VIC) for land surface processes, and the enhanced Atmospheric Wave Boundary Layer Model (e-AWBLM) for atmospheric forcing.

There is also a potential for physics-based coupling across multiple environmental domains, where wave-circulation models operate in tight coupling with frequent information exchange, while ocean-river boundaries facilitate less frequent but equally critical data transfer between hydrodynamic and terrestrial routing models (Tanim et al., 2024). These comprehensive coupling methods allow for the representation of complex nonlinear interactions where storm surge affects river discharge through changes in water levels, while river flows influence coastal circulation patterns, demonstrating substantial accuracy improvements over sequential approaches, especially for predicting extreme events and mapping inundation.

Despite the extensive development of physics-based coupled systems, the application of hybrid feedback models that integrate physics-based and statistical approaches remains limited in compound flood modeling literature. Basically, all the physics-based coupled systems mentioned above, while sharing the bidirectional information exchange feature of feedback hybrid models, differ fundamentally in that they connect models within the same modeling paradigm rather than linking physics-based and statistical approaches. While physics-based coupled systems provide a more realistic simulation of compound dynamics than sequential models, they are often computationally expensive, limiting their application for real-time forecasting or probabilistic hazard assessment. Moreover, physics-based models must make numerous assumptions and parameterizations about the conditions of the simulation domain. In some cases, these assumptions and parameterizations can introduce errors in the simulation. Hybrid feedback models could reduce the computational burden by replacing numerical solvers with trained statistical models and may be able to better represent parameterized processes by learning directly from observational data. However, the scarcity of feedback hybrid models reflects the technical challenges inherent in establishing bidirectional communication between fundamentally different modeling paradigms. The few existing implementations show the potential for merging mechanistic understanding with data-driven efficiency through iterative coupling strategies. (Li, 2021) developed one of the few documented hybrid feedback frameworks by coupling a sequence-to-sequence LSTM rainfall-runoff model with the ADCIRC solver to simulate compound flooding in Houston's Brays Bayou and Galveston Bay, where the LSTM provides updated discharge hydrographs to ADCIRC at each coupling interval while ADCIRC returns surge-driven water-level changes that are immediately fed back into the neural network, creating a





continuous cycle where the data-driven hydrologic predictor and physics-based hydrodynamic model continually adjust one another throughout event evolution.

Recent advances in physics-informed machine learning (Raissi et al., 2019) and coastal-oceanic data assimilation techniques (Canizares et al., 1998; Heemink, 1986; Verlaan and Heemink, 1997) have catalyzed the emergence of hybrid feedback approaches that seamlessly integrate physics-based and statistical modeling components. Data assimilation

frameworks exemplify one pathway for hybrid feedback integration, where statistical algorithms continuously incorporate observational data to update physics-based model states while mechanistic predictions guide the statistical processing of new observations. (Muñoz et al., 2022a) demonstrated this approach by integrating a Bayesian data assimilation scheme with Delft3D-FM for compound flood prediction, where the coupled coastal-inland flood model operates in tandem with an ensemble Kalman filter that ingests real-time water level observations every few hours, creating continuous information

exchange where model forecasts receive statistical updates from the data-assimilation module while corrected state estimates feed back into the hydrodynamic simulation (Figure B1).

Physics-informed machine learning (PIML) constitutes an alternative pathway for hybrid feedback coupling through the direct incorporation of physical principles into statistical learning frameworks. This methodology ensures data-driven predictions maintain consistency with fundamental physical laws while capitalizing on ML's computational advantages.

(Donnelly et al., 2024) introduced an approach by adding mass-conservation constraints into neural network optimization functions, allowing networks to replicate flood solver behavior while maintaining physical consistency. This means the neural net "learns" the flood dynamics while respecting key physics, effectively coupling the statistical model with the governing coastal dynamics during training. Building on this foundation, (Radfar et al., 2025) expanded the constraint framework to include full shallow water dynamics, covering mass and momentum conservation, and developed a

generalizable emulator that maintains physical consistency in unseen scenarios while outperforming unconstrained data-driven alternatives at compound flood peaks.

**3.3 Ensemble Modeling**

Ensemble hybrid models integrate multiple process-specific modeling systems that each simulate the same phenomenon but often employ differing methodologies, such as purely physics-based, statistical, or ML approaches. Rather than exchanging

boundary conditions or state variables in real time, each model separately produces an output (e.g., a river flow forecast, a spatial inundation extent, and a time series of water levels) based on its own domain assumptions, forcing data, and internal processes. Although ensemble schemes do not involve the two-way exchange of state variables or iterative feedback, they remain "hybrid" in the sense that they unite multiple independently validated solutions into a single framework for final decision-making or risk assessment. As with sequential or feedback categories, the defining trait is that all the models

involved can be executed in isolation; the difference lies in how and when their outputs are combined. This approach offers





advantages for uncertainty quantification, particularly for structural uncertainty that arises from limitations in model design and underlying assumptions (Abbaszadeh et al., 2022).

A typical ensemble framework (see Figure 5 and Figure C1) combines physics-based and data-driven models that target the same variable, such as peak flood height, to leverage their complementary strengths. The physics-based component uses

high-fidelity solvers (e.g., ADCIRC, Delft3D-FM) to resolve the governing dynamics, while the statistical or ML component provides fast surrogates (e.g., Gaussian-process models, PIML) that can extrapolate patterns to ungauged locations or generate rapid predictions. Each model independently delivers its own estimate of the target variable using its distinct methodology. Instead of exchanging information iteratively, ensemble methodologies combine or "pool" their final outputs through a dedicated ensemble algorithm. Ensemble selection can be guided by expert judgment, agent-based models, or by

applying weighted or Bayesian combinations based on historical performance or evaluation metrics. Figure C1 presents a detailed example of ensemble hybrid models where an ensemble of a physics-based model and a Bayesian analog model is used to generate a precipitation forecast.

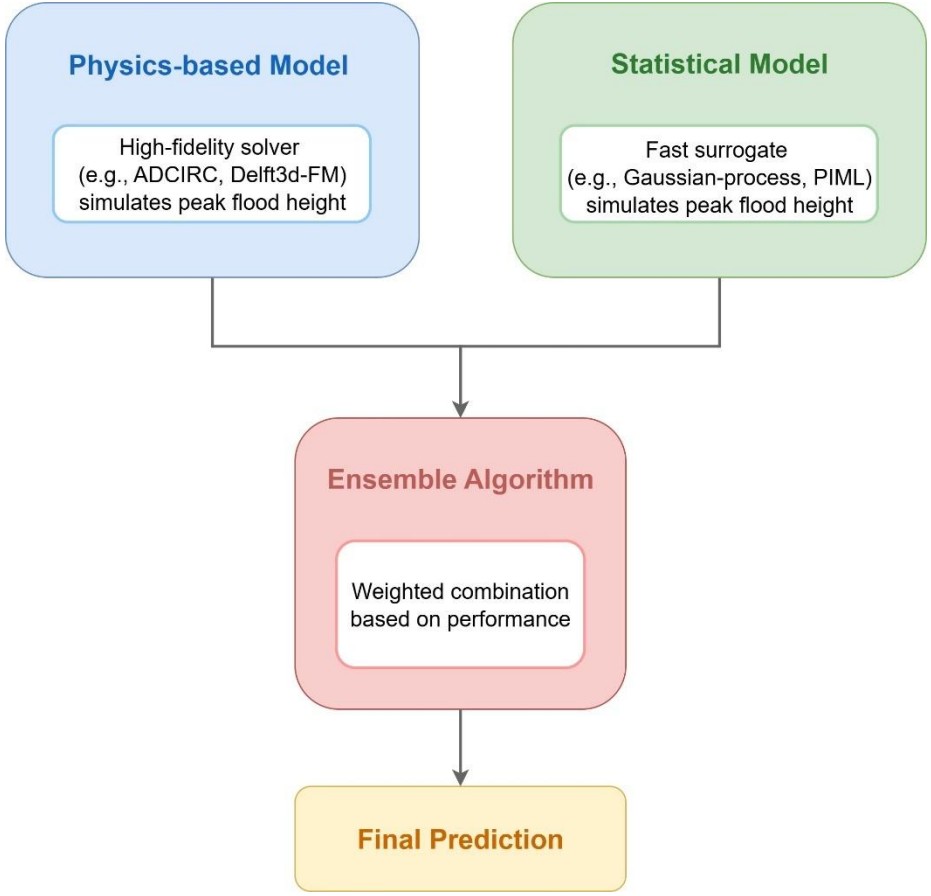





**Figure 5. Schematic illustration of an ensemble hybrid modeling framework showing the integration of different modeling approaches.** The physics-based model component simulates peak flood height, while the statistical model component analyzes historical patterns and relationships. An ensemble algorithm combines outputs from both models through a weighted combination based on performance metrics. The final prediction represents the optimally weighted combination of model predictions, demonstrating how ensemble hybrid methods leverage the strengths of different modeling approaches, where complete solutions are merged rather than coupled.

The ensemble approach offers several advantages for compound flood modeling. By integrating different modeling paradigms, it can capture a broader range of uncertainties and leverage the unique strengths of each approach. The ensemble framework can also adapt to different conditions by adjusting the weights assigned to each model based on its performance in similar situations. For instance, during extreme events, the framework might give more weight to physics-based predictions, while statistical models might receive higher weights during more typical conditions. The key distinction of ensemble hybrid models lies in their focus on combining complete, independent solutions rather than coupling model components during simulation. This approach can be particularly valuable when different modeling paradigms offer complementary insights into the same phenomenon. Through a careful combination of these diverse approaches, ensemble hybrid frameworks can provide more robust and reliable predictions while maintaining the independence and specialized capabilities of each modeling component.

## 4. Decision framework and practical insights

### 4.1 Comparative strengths and limitations

Each hybrid physical–statistical modeling approach offers distinct advantages and faces specific challenges when applied to compound flooding assessment:

**Sequential hybrid models** provide an accessible entry point for practitioners with limited computational resources. Their straightforward implementation makes them ideal for areas where the influence between processes is predominantly unidirectional (e.g., upstream to downstream). However, they do not capture feedback mechanisms between systems, potentially missing important interactions. Sequential approaches are most appropriate when:

- Interaction between physical processes is primarily one-directional;
- Computational resources are limited;
- The study area has clear physical boundaries between domains; and
- Rapid assessment is prioritized over capturing all physical interactions.

**Feedback hybrid models** excel at representing the dynamic interactions between environmental processes, making them particularly valuable in coastal transition zones where river discharge, tidal effects, and storm surge interact in complex ways. These models can capture nonlinear feedback critical for the accurate prediction of water levels during extreme events.



However, they require significantly greater computational resources and expertise to implement and maintain. Feedback approaches are most suitable when:

- Bi-directional processes significantly influence outcomes;
- Computational resources are substantial;
- Physical interactions cross domain boundaries frequently; and
- Accurate representation of system dynamics is prioritized.

**Ensemble hybrid models** enhance uncertainty handling by leveraging multiple modeling paradigms. They excel at balancing the strengths of different approaches and can adapt to various conditions by adjusting model weights. While they may require running multiple complete models, they often allow parallel execution, potentially reducing overall computation time. Ensemble approaches are particularly valuable when:

- Handling uncertainty is a priority;
- Multiple modeling methodologies are available;
- Historical performance data exists for model validation; and
- The target variables benefit from diverse prediction methods.

## 4.2 Model selection framework

Selecting an appropriate hybrid model requires not only a clear typology of coupling strategies but also an understanding of the typical compound hazard scenarios for which these models are applied. In typical coastal settings various combinations of coastal oceanic and terrestrial drivers of flooding combine to produce specific categories of compound flooding (Green et al., 2025). These categories provide the context in which modelers decide whether a sequential, feedback, or ensemble design is most appropriate. Table 1 links each scenario to representative model pairs, their classification within the hybrid 425 typology, and studies that demonstrate their application. It is important to note that a large body of existing literature on compound flood modeling (Table 1 and Section 3) falls under the category of sequential hybrid models. The limited number of examples in the other categories primarily reflects their greater technical challenges—such as the need for continuous data assimilation, computationally intensive coupling, consistent ensemble integration, and ensuring conservation of physical laws in ML models. As computational tools and data streams continue to advance, broader adoption of these less common 430 but powerful hybrid approaches is expected.



**Table 1. Representative hybrid modeling approaches for coastal compound flooding.** The table summarizes examples of coastal compound flood hazard scenarios, their hybrid model components, typology classification, and studies that illustrate each modeling approach.

| Typical coastal compound hazard scenario | Hybrid model components | Typology class | Representative study |
|---|---|---|---|
| River-surge/tide (Fluvial + coastal) | Gaussian-copula Bayesian network (river discharge and surge combinations) + HEC-RAS (model water level) | Sequential | (Couasnon et al., 2018) |
| Rainfall-surge/tide (Pluvial + coastal) | Statistical rainfall generator (rainfall) + ADCIRC (surge/tide) | Sequential | (Bilskie et al., 2021; Gori and Lin, 2022) |
| River + Rainfall + surge/tide (Fluvial + pluvial + coastal) | Trivariate copula event generator (rainfall, river discharge, surge peaks) + 1D river network hydrodynamic model (compound inundation) | Sequential | (Wang et al., 2025) |
| | LSTM model (rainfall-runoff) + ADCIRC (surge/tide) | Feedback | (Li, 2021) |
| | Delft3D-FM (compound flood dynamics) + Ensemble Kalman Filter (water level data assimilation) | Feedback | (Muñoz et al., 2022a) |
| Sea-level/surge-driven groundwater rise (Coastal + groundwater) | Flood frequency analysis (Bayesian hierarchical model) + groundwater modeling (MODFLOW) | Sequential | (Habel et al., 2020) |


To further guide practitioners in selecting the most appropriate hybrid physical–statistical modeling approach for their specific needs, we propose a decision framework (Figure 6). This framework addresses key considerations, including physical system characteristics, study objectives, and resource constraints.





**Figure 6. Decision tree for hybrid model selection in compound flood modeling.** The framework provides a structured approach to selecting appropriate modeling strategies based on key decision criteria. Each terminal node represents a recommended modeling approach with associated key considerations.

The decision process begins with an assessment of the physical system under study, particularly whether modeling interactions between different flooding mechanisms is critical. If not, a single-domain model may be sufficient. When interactions are important, the next consideration is whether both a physics-based model and a statistical/data-driven model are available. Next, for systems where bi-directional interactions are significant (e.g., river discharge affecting coastal water levels while coastal surge simultaneously impacts riverine processes), the framework directs practitioners to evaluate their computational resources. When adequate resources are available, feedback models are recommended to capture the dynamic interactions between domains. However, if computational constraints exist, sequential models serve as a practical alternative that preserves some representation of system interactions while requiring fewer resources.



For systems where interactions are predominantly unidirectional, the framework guides practitioners to consider whether the component models will run in parallel and have their outputs combined. A "Yes" directs users to ensemble models, which run the physics and statistical elements independently and merge their results afterward, offering advantages through structural flexibility and the ability to leverage complementary strengths of different approaches. A "No" keeps the workflow in the sequential category, where outputs from one component feed directly into the other during execution. Sequential approaches provide an efficient solution that maintains sufficient accuracy while requiring relatively modest computational resources.

Each modeling approach is accompanied by key considerations that highlight its specific strengths and applications. Sequential models excel in situations requiring rapid assessment with clear domain boundaries and predominantly one-way processes. Ensemble models leverage structural diversity and independent execution capabilities while handling uncertainty effectively. Feedback models provide the highest accuracy for systems with complex physics and cross-domain interactions, but demand greater computational resources. By following this decision framework, practitioners can systematically evaluate their specific needs and constraints to select the most appropriate hybrid modeling approach for compound flooding assessment.

Model choices directly shape what we can learn about compound flooding. Statistical frameworks such as copulas, Bayesian joint probability methods, and event-based sampling approaches (Couasnon et al., 2018; Gori et al., 2020; Gori et al., 2022; Cho et al., 2023) improve our understanding of dependence structures and allow exploration of extreme but physically plausible driver combinations, thereby reducing sampling uncertainty compared to using only historical events. At the process level, selecting particular hydrodynamic cores (e.g., ADCIRC, SWAN, Delft3D) or coupling strategies (Luettich and Westerink, 2004; Booij et al., 1999; Lawler et al., 2016) determines how well key interactions such as tide–surge–wave coupling or river backwater effects are represented. Feedback-oriented hybrids, particularly data assimilation approaches, enhance our ability to track evolving flood states while uncovering sensitivities in surge–river interactions that one-way simulations often miss (Canizares et al., 1998; Heemink, 1986; Muñoz et al., 2022a). At broader scales, multi-model chains and storyline frameworks demonstrate how compound hazard amplification emerges under climate variability and sea-level rise, highlighting where system dynamics are most uncertain and where resilience measures may be most needed (Gori and Lin, 2022; Muñoz et al., 2022b; Eilander et al., 2023). In all these cases, model choice is not just a technical decision but a scientific one, influencing both quantitative estimates and qualitative understanding of how compound floods arise.

### 4.3 Implementation of workflow guidelines

To assist practitioners in selecting and implementing appropriate hybrid modeling approaches for compound flooding, we propose the following workflow:



1. **Assessment of physical processes**: Identify the key physical processes contributing to compound flooding in the study area and their potential interactions. Determine whether these interactions are primarily one-directional or involve significant feedback.

2. **Evaluation of available resources**: Consider computational capacity, data availability, expertise, and time constraints. Feedback models typically require more resources than sequential models, while ensemble models may benefit from parallel computing infrastructure.

3. **Accuracy requirements assessment**: Define the required level of accuracy for the modeling objectives, considering the intended application (e.g., real-time forecasting vs. long-term planning, regulatory compliance vs. research purposes). Determine acceptable uncertainty ranges and error tolerances that will influence the complexity and type of hybrid approach needed.

4. **Selection of an appropriate hybrid approach**: Based on the resource assessment (step 2) and accuracy requirements (step 3), select the most suitable modeling framework:

   - For areas with clear upstream-downstream relationships and limited feedback, consider sequential approaches.

   - For regions with complex interaction zones where feedback significantly affects outcomes, prioritize feedback approaches.

   - When multiple modeling paradigms show complementary strengths or when uncertainty quantification is critical, employ ensemble methods.

5. **Interface design**: Develop appropriate interfaces between model components, considering data format compatibility, spatial and temporal resolution differences, and transformation requirements.

6. **Validation strategy**: Implement comprehensive validation protocols for both individual components and the integrated system, using observational data where available and sensitivity testing to evaluate model behavior.

7. **Uncertainty analysis**: Assess and communicate uncertainties associated with each component and the integrated system, particularly focusing on how uncertainties propagate through model coupling.

## 5. Summary and implications

The hybrid physical–statistical modeling frameworks presented in this paper demonstrate significant capabilities for simulating compound flooding processes. These approaches include physics-based models that simulate physical phenomena under investigation, as well as the nonlinear interactions among the variables involved, and data-driven statistical models that capture the interdependencies of drivers and their associated impacts. By integrating cross-paradigm models, each contributing specialized capabilities and validated methodologies, these frameworks enable more comprehensive and robust simulations than single-model approaches. The strength of hybrid frameworks lies in their ability to preserve the





independence of each component while creating integrated solutions that capture the complex interactions inherent in compound flooding phenomena. Table 2 summarizes the key characteristics of hybrid modeling frameworks, highlighting their core definition, cross-paradigm, process-specific roles, and nature of coupling. These fundamental aspects provide a foundation for understanding the essential features that define these integrated modeling systems, regardless of the specific implementation approach chosen.

**Table 2. Key characteristics of hybrid modeling frameworks for compound flood simulation.** The table presents fundamental aspects of hybrid models, including their core definition, cross-paradigm, process-specific roles, and coupling approaches, highlighting the essential features that define these integrated modeling systems.

| Feature / Criterion | Characteristics of hybrid models |
|---|---|
| Core definition | Two (or more) process-specific models from different modeling paradigms (e.g., physics-based and statistical/data-driven), each capable of simulating its own subdomain, working together within a unified modeling framework. Each component model maintains its full functionality within its respective domain. |
| Cross-paradigm and process-specific | Each component targets one or more flood drivers or subprocesses (e.g., atmospheric forcing, watershed runoff, coastal surge, or wave dynamics) using the methodology best suited to that role (governing-equation solver, copula sampler, PIML, EnKF, etc.). Components can be developed, calibrated, and validated independently, yet their complementary outputs are combined to form the integrated multi-physics simulation. |
| Nature of coupling | Can be implemented as sequential (one-way), feedback (two-way), or ensemble (output-level). |
| Integration level | Integration may occur during simulation (e.g., iterative feedback of wave radiation stresses to a surge model) or after each model run (ensemble). Sequential implementations maintain a one-way flow of information between components, while feedback approaches enable two-way exchanges. |
| Resulting scope | Offers a more robust and holistic simulation of compound hazards by leveraging each model's specialized capabilities. Addresses complex interactions between subdomains (river discharge, coastal surge, waves, etc.) by combining domain-specific expertise into a comprehensive framework. |
| Computational considerations | Varies by implementation category. Sequential approaches typically require less coordination than feedback models, while ensemble methods allow for independent execution followed by a strategic combination of outputs. All approaches balance computational demands with the benefits of multi-model integration. |





| Feature / Criterion | Characteristics of hybrid models |
| --- | --- |
| Key advantages | Enables more accurate representation of compound phenomena than single-model approaches. Leverages specialist expertise in different domains. Provides a framework for quantifying structural uncertainty through model diversity. Enhances flexibility in representing complex physical interactions. |
| Implementation challenges | Requires careful interface design between components. It may involve data transformation between different spatial or temporal scales. Requires expertise across multiple modeling domains. Necessitates comprehensive validation to ensure component interactions are physically realistic. |

Whether implemented through sequential chains, feedback loops, or ensemble methods, hybrid frameworks offer flexible and powerful approaches to address the challenges of compound flood modeling. Sequential hybrid implementations provide straightforward yet effective ways to link cross-paradigm models, allowing each to contribute its specific strengths
to the overall simulation. Feedback-based approaches enable more sophisticated interactions between components, capturing complex physical processes and their interdependencies. Ensemble methods leverage multiple independent solutions to better characterize uncertainty and improve prediction reliability. The flexibility to choose between sequential, feedback, or ensemble implementations allows modelers to balance complexity, computational cost, and accuracy requirements based on specific needs and resources.

Looking forward, the continued development and refinement of hybrid modeling approaches hold great promise for advancing the ability to understand, predict, and manage compound flooding risks. The potential research directions are:

- **Standardized coupling interfaces**: Development of standardized protocols and software interfaces would facilitate more efficient model integration and comparison across different hybrid implementations.
- **Adaptive coupling strategies**: Research into dynamic coupling approaches that adjust information exchange
frequency based on event characteristics could optimize computational efficiency.
- **Machine learning integration**: Exploring how ML techniques can complement physics-based models within hybrid frameworks, particularly for parameter optimization, uncertainty quantification, and real-time prediction.
- **Multi-scale coupling**: Advancing methods to bridge spatial and temporal scale differences between model
components while preserving critical physical processes.
- **Operational implementation**: Transitioning hybrid modeling approaches from research to operational forecasting systems, including strategies for reducing computational demands (e.g., deriving vehicle-specific flood-closure thresholds as in (Maghsoodifar et al., 2025)).



As computational capabilities expand and new modeling techniques emerge, the potential for innovative hybrid implementations will grow. Nevertheless, the systematic framework and decision guidance presented in this paper provide a foundation for selecting and implementing appropriate hybrid modeling approaches for compound flooding assessment. By adopting a consistent terminology and structured selection process, practitioners can more effectively identify suitable methodologies, anticipate implementation challenges, and develop robust solutions for understanding and predicting compound flooding in diverse coastal environments.

## Appendix A: Technical implementation example of sequential hybrid models

An example of a sequential hybrid workflow is given by (Gori and Lin, 2022), who evaluated future compound flood hazard for the Cape Fear Estuary, North Carolina. Their system links several process-specific models in one direction. A statistical synthetic tropical cyclone (TC) generator first produces thousands of storm tracks. Each track then drives two independent models: a TC-Rainfall (TCR) convective model, which yields gridded rainfall, and an ADCIRC storm-tide model. Peak rainfall, peak surge, and their timing are passed to a JPM-OS-BQ joint probability module, which picks a representative subset of events. The selected rainfall fields are fed into HEC-HMS to create watershed inflow hydrographs. These hydrographs, together with the selected storm-tide series, serve as boundary conditions for a high-resolution HEC-RAS 1D/2D inundation model. Information moves only forward—statistics to physics, upstream physics to downstream physics—while each component retains its full, native capabilities. This one-way transfer of information exemplifies a sequential hybrid modeling framework (Figure A1).



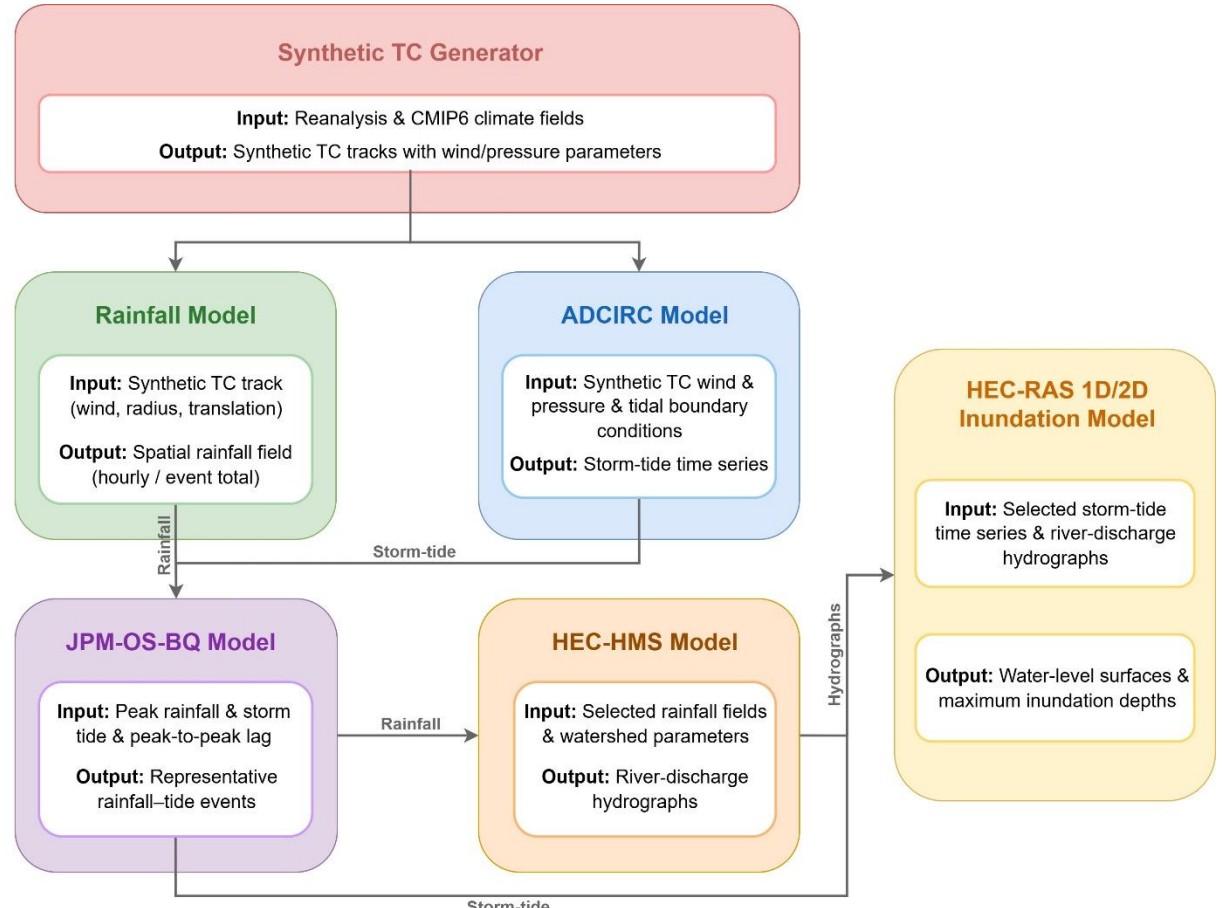

**Figure A1. Technical implementation of a sequential hybrid framework adapted from (Gori and Lin, 2022) for projecting compound flood hazard in the Cape Fear Estuary, NC.** A statistical Synthetic-TC generator (top) converts reanalysis and CMIP6 climate fields into thousands of synthetic TC tracks with intensity-size parameters. Two physics-based engines then operate independently on each track: a convective rainfall model (left) produces gridded storm-rainfall fields, while an ADCIRC hydrodynamic model (right) simulates coastal storm-tide time series. Their peak rainfall, peak surge, and time lag feed a statistical JPM-OS-BQ selector that distils a representative subset of rainfall–tide events (purple). Those selected rainfall fields drive a regional HEC-HMS hydrology model, yielding river-discharge hydrographs that, together with the selected storm-tide series, serve as boundary conditions for a high-resolution HEC-RAS 1D/2D inundation model (gold), which maps water-level surfaces and maximum depths. Grey arrows denote the one-way flow of information, illustrating the characteristic of sequential hybrid approaches where each component maintains complete simulation capabilities in its respective domain while information flows predominantly in one direction.



**Appendix B: Technical implementation example of feedback hybrid models**

Figure B1 illustrates the feedback hybrid model provided by (Muñoz et al., 2022a) in their Gulf Coast compound flood forecast system. Two components operate continuously and exchange information during each assimilation window. The
575 Delft3D-FM hydrodynamic model predicts water level and current fields based on wind, pressure, tide, and river inflow forcing, providing an ensemble forecast for the next model step. Those forecast fields, along with real-time water level observations and their error statistics, are processed by an Ensemble Kalman Filter (EnKF). The EnKF combines the model prediction with observations, producing a corrected state ensemble (i.e., updated water levels and velocities) that immediately replaces Delft3D-FM's initial conditions before the solver resumes. This two-way loop repeats every few hours,
so surge-river interactions reflected in the gauges feed directly back into the next hydrodynamic calculation. They showed that the feedback cycle reduces water level errors by up to 40% compared with a one-way run, especially during peak surge when coastal backwater effects and river inflows interact most strongly.

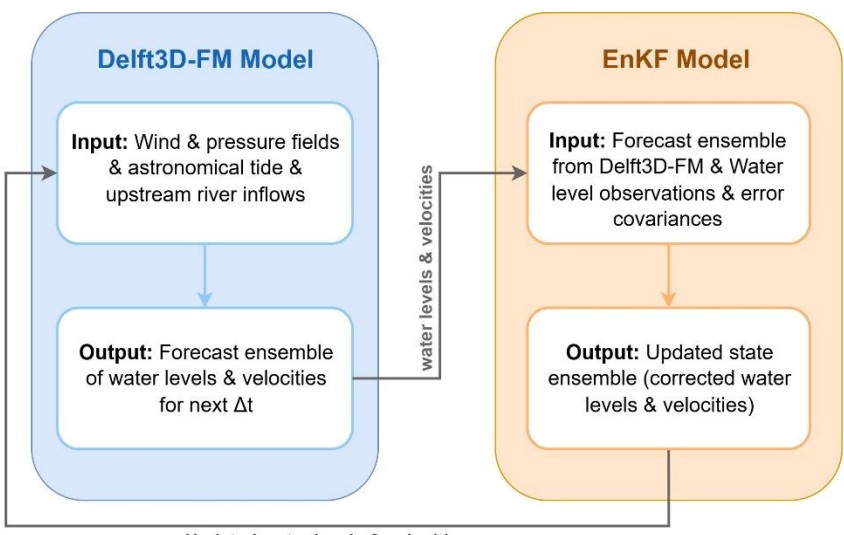

**Figure B1. Technical implementation of a feedback hybrid modeling framework based on (Muñoz et al., 2022a)'s approach for**
**simulating compound flooding along the northern Gulf Coast.** The framework consists of two process-specific models: Delft3D-FM (hydrodynamic coastal–riverine model) and an Ensemble Kalman Filter (EnKF) (statistical data assimilation module). Coloured boxes indicate the two modeling domains. Arrows represent information flow between the models at each assimilation cycle, with water level and velocity fields exchanged as indicated. Each model maintains complete simulation capabilities in its respective domain while exchanging essential data with the other model to capture compound flood dynamics through continuous feedback coupling.



## Appendix C: Technical implementation example of ensemble hybrid models

A practical example of ensemble hybrid modeling can be found in (Madadgar et al., 2016)'s study of seasonal precipitation forecasting in the southwestern United States (Figure C1). In their implementation, two process-specific models, NMME and a Bayesian analog model, were combined through an ensemble approach to improve prediction skill. Their framework integrates two complete models: NMME as the physics-based component with 99 ensemble members providing dynamical simulations, and a Bayesian analog model as the statistical component incorporating teleconnection indices (PDO, MEI, and AMO). Rather than operating iteratively, these components function independently, and their outputs are overlaid for combination through an Expert Advice algorithm. This algorithm applies a weighted ensemble combination based on each model's historical performance, effectively pooling their predictions to generate a final precipitation forecast. By comparing and consolidating multiple approaches to the same hazard, this ensemble framework demonstrates how both physically based simulations and statistical approaches can be leveraged to improve prediction skill for complex phenomena like seasonal precipitation forecasting.

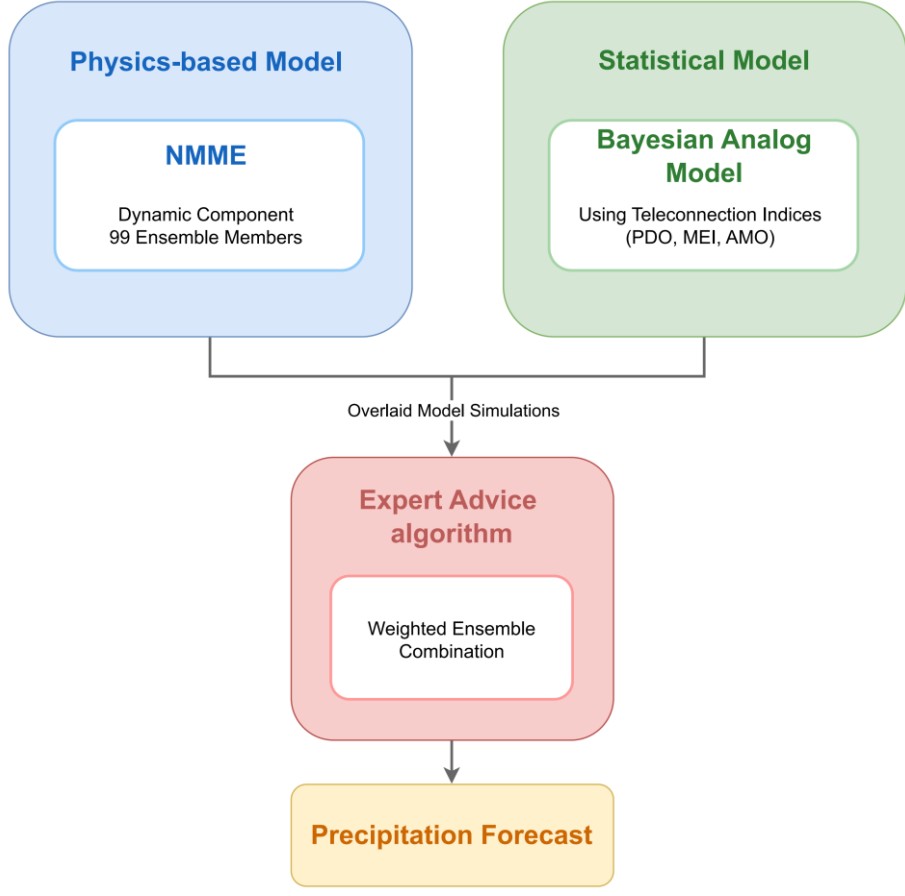



**Figure C1. Technical implementation of an ensemble hybrid modeling framework based on (Madadgar et al., 2016)'s approach for seasonal precipitation and drought prediction in the southwestern United States.** The physics-based model component (blue box) employs NMME as a complete dynamical simulator with 99 ensemble members incorporating atmospheric-ocean interactions. The statistical model component (green box) uses a process-specific Bayesian analog approach that generates complete predictions using teleconnection indices (PDO, MEI, AMO). Gray arrows indicate information flow to the Expert Advice algorithm (red box), which performs a weighted ensemble combination based on historical performance metrics. This ensemble hybrid approach enables independent execution of each modeling component while strategically combining their outputs through a unified framework to leverage the strengths of both physical and statistical predictions, characteristic of ensemble hybrid methods, where complete solutions are merged rather than coupled. The final precipitation forecast (yellow box) represents the optimally weighted combination of all model predictions.

*Author contributions*. SR and HM conceptualized the study. SR wrote the original draft and prepared the visualizations. HM, DFM, AG, FD, NL, and AA provided critical feedback, revisions, and review of the manuscript.

*Competing interests*. The contact author has declared that none of the authors has any competing interests.

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
