# Peer review of "Towards a typology for hybrid compound flood modeling"

_EGUsphere, 2025_

## Author Comment (AC1)

**Hydrology and Earth System Sciences**

Notes on revision made to manuscript egusphere-2025-4623

**Reviewer 1**

**Reviewer Comment 1.1 —** I would like to compliment the authors on a very thorough and useful paper. I think there is great value in this very thorough review, and I think output such as Figure 6 can be very helpful. I did find the paper very dense which affects its readability; I have included some feedback below which maybe helps in addressing this.

**Reply:** We thank the reviewer for taking the time to read the manuscript and provide constructive feedback. In response to the comments, we have revised the paper to improve clarity and readability by adding a table of key definitions (Table 1), introducing a new subsection on the roles of the statistical/ML component in hybrid models (Section 4.2), and expanding the discussion of future directions, including links to socio-hydrological modeling (Section 5). Detailed responses to all comments are provided below.

**Reviewer Comment 1.2 —** There are a lot of different definitions discussed including their synonyms, and I wondered if adding a box with key definitions may help the reader. (eg two-way (or loosely) coupled models and tightly coupled models, etc.) Or maybe it can be added to existing figures such as figure 2.

**Reply:** As per your suggestion, we have added Table 1 with definitions for 11 technical terms.

| Term | Definition |
|---|---|
| Compound flooding | Flooding arises from a combination of multiple drivers—such as storm surge, river discharge, rainfall, waves, or tides—whose interactions may amplify the hazard (Moftakhari et al., 2017; Wahl et al., 2015). |
| Process-based model | A numerical model that solves governing physical equations (e.g., Navier–Stokes, shallow-water equations) to simulate hydrodynamic, hydraulic, atmospheric, or surface-runoff processes. |
| Statistical model | A model that represents system behavior through probability distributions, empirical relationships, or data-driven statistical structures rather than directly solving physical equations. |
| Coupling | The integration of two or more models so that the output of one drives (and sometimes is also driven by) another, allowing different environmental domains to be represented in one workflow. |
| Two-way (loosely) coupled model | Two independent models run separately but exchange information in both directions at set intervals. The codes remain distinct and are called iteratively, so feedback is captured but with lower temporal granularity than in tight coupling (Santiago-Collazo et al., 2019). |

**Hydrology and Earth System Sciences**
Notes on revision made to manuscript egusphere-2025-4623

| Term | Definition |
|---|---|
| Tightly (fully) coupled model | Model components are merged into a single executable or solver framework and share state variables at every (or very fine) time-step, ensuring high-frequency, bidirectional feedback (Santiago-Collazo et al., 2019). |
| Bias correction | Statistical adjustment of model outputs to correct systematic deviations between modeled and observed data (Maraun and Widmann, 2018). |
| Data assimilation | Statistical methods that blend model forecasts with observations to estimate the evolving system state (Moradkhani et al., 2005). |
| Uncertainty quantification | Identification, characterization, and propagation of uncertainties arising from forcings, parameters, initial states, assumptions and model structure—and strategies to reduce them (Abbaszadeh et al., 2022; Muñoz et al., 2024). |
| Copula | A multivariate distribution that couples given marginals into a joint distribution, allowing flexible modelling of dependence (including tail behavior) (Nelsen, 2006). |
| Physics-informed machine learning | Machine-learning techniques in which physical laws (e.g., partial differential equations, conservation principles) are imposed as soft constraints in the loss function or as hard constraints that are built directly into the model architecture, yielding data-efficient, physically consistent surrogates (Raissi et al., 2019). |

**Reviewer Comment 1.3 —** In sections 3.1 and 3.2, I was wondering about the "statistical component" and whether it would help to include a classification of the different roles or function of the statistical component within the different types of coupling. Clarifying the different roles of the statistical component within a hybrid model may help users to classify their specific hybrid approach?

**Reply:** In the revised version, we added a new section, "4.2 Roles of the statistical component in hybrid models," to summarize the different functions of the statistical component within a hybrid model:

> *4.2 Roles of the statistical component in hybrid models*
>
> *In hybrid modeling frameworks, the statistical/ML component plays different roles depending on the coupling strategy and the compound flood problem being addressed. Although Section 3 introduces statistical tools within each hybrid category, the functional purpose of the statistical block varies considerably across sequential, feedback, and ensemble designs. Across these paradigms, the statistical/ML block typically performs one or more of the five roles listed in Table 2:*

**Hydrology and Earth System Sciences**
Notes on revision made to manuscript egusphere-2025-4623

**Table 1. Roles of the statistical/ML component across hybrid modeling pathways**

| Role of statistical/ML component | Description | Sequential | Feedback | Ensemble |
|---|---|---|---|---|
| (1) Boundary/driver generator | Supplies synthetic or real-time forcings (e.g., rainfall, discharge, surge) as inputs to a physics-based solver. | ✓ | — | (✓)* |
| (2) Scenario sampler/event-catalogue builder | Draws joint extremes (e.g., via copulas/multivariate methods) to expand beyond historical records. | ✓ | (✓)* | (✓)* |
| (3) Physics surrogate | Provides a computationally efficient surrogate for an otherwise expensive physics-based model (e.g., rainfall–runoff, PIML hydrodynamics). | — | ✓ | (✓)* |
| (4) State updater | Assimilates observations or corrects model states during runtime; characteristic of two-way feedback systems. | — | ✓ | (✓)* |
| (5) Ensemble aggregator | Combines independent predictions using performance-, uncertainty-, or cost-based weighting. | — | — | ✓ |

\* Parentheses indicate roles that may appear in some implementations but are not core to the pathway.

*Mapping these roles helps clarify how the statistical component contributes to the broader simulation framework and provides a consistent basis for classifying hybrid approaches. As summarized in Table 2, sequential hybrids typically rely on roles (1) and (2); feedback hybrids make use of roles (3) and (4) (and occasionally role 2); and ensemble hybrids center on role (5) while potentially incorporating roles (1)–(4) within individual ensemble members. This functional perspective highlights that the statistical or ML component is not a single construct but a spectrum of tasks that complement the physics engine in different ways.*"

**Reviewer Comment 1.4 —** Finally, I was wondering if there would be premise in adding a small discussion about linking (the framework of) compound flood modelling to more socio-hydrological models that capture not only compound flood / hydrological dynamics but also their interactions with people; eg https://gmd.copernicus.org/articles/16/2437/2023/?.

**Reply:** We added a new paragraph to Section 5 to emphasize this potential future pathway:

"*Emerging socio-hydrological and agent-based frameworks also offer opportunities for expanding compound flood modeling beyond physical drivers alone. These models explicitly simulate the feedbacks between human decisions and hydrological responses across a wide range of spatial scales, from large-scale agent-based systems, where millions of agents interact dynamically with soil moisture, groundwater, reservoirs, and routing processes (De Bruijn et al., 2023), to household-*

*level adaptation models (Haer et al., 2020), catchment-scale frameworks that couple agent behavior with calibrated runoff responses (Sousa et al., 2025), behavior-aware reservoir-operation schemes (Gautam et al., 2025), and more generic socio-hydrological agent-based platforms developed for integrated water management applications (Lillo-Saavedra et al., 2024). Integrating compound flood frameworks with such socio-hydrological models could allow future studies to capture not only the multivariate flood physics but also how human behavior, adaptation, exposure, and decision-making co-evolve with compound flood hazards under changing climate and socioeconomic conditions.*"

***References:***

De Bruijn, J. A., Smilovic, M., Burek, P., Guillaumot, L., Wada, Y., & Aerts, J. C. (2023). GEB v0. 1: a large-scale agent-based socio-hydrological model–simulating 10 million individual farming households in a fully distributed hydrological model. *Geoscientific Model Development*, *16*(9), 2437-2454.

Haer, T., Husby, T. G., Botzen, W. W., & Aerts, J. C. (2020). The safe development paradox: An agent-based model for flood risk under climate change in the European Union. *Global Environmental Change*, *60*, 102009.

Gautam, S., Park, S., Yu, D. J., Garcia, M., Sivapalan, M., & Shin, H. C. (2025). Homo juridicus, homo heuristicus, and homo anticipans: A sociohydrological study of operator behavior and flood-drought tradeoffs in reservoirs. *Water Resources Research*, *61*(11), e2024WR039447.

Lillo-Saavedra, M., Velásquez-Cisterna, P., García-Pedrero, Á., Salgado-Vargas, M., Rivera, D., Cisterna-Roa, V., ... & Gonzalo-Martín, C. (2024). Socio-Hydrological Agent-Based Modeling as a Framework for Analyzing Conflicts Within Water User Organizations. *Water*, *16*(22), 3321.

Sousa, D. S., Silva, E. P., de MA Alves, C., Minoti, R. T., & Vergara, F. E. (2025). Coupling data-driven agent-based and hydrological modelling to explore the effect of collective water allocation strategies in environmental flows. *Journal of Hydrology*, *652*, 132670.

**Reviewer Comment 1.5 —** Some literature suggestions. Tilloy et al 2019, Mishra et al 2022.

**Reply:** As per your suggestion, we cited these two papers in the Introduction:

- Tilloy, A., Malamud, B. D., Winter, H., & Joly-Laugel, A. (2019). A review of quantification methodologies for multi-hazard interrelationships. *Earth-Science Reviews*, *196*, 102881.

**Hydrology and Earth System Sciences**
Notes on revision made to manuscript egusphere-2025-4623

- Mishra, A., Mukherjee, S., Merz, B., Singh, V. P., Wright, D. B., Villarini, G., ... & Stedinger, J. R. (2022). An overview of flood concepts, challenges, and future directions. *Journal of hydrologic engineering*, *27*(6), 03122001.

---

## Author Comment (AC2)

**Hydrology and Earth System Sciences**

Notes on revision made to manuscript egusphere-2025-4623

**Reviewer 2**

**Reviewer Comment 2.1 —** I would like to compliment the authors for a well-organized, and clearly written manuscript that addresses a timely and underexplored topic: the classification of hybrid compound flood modeling frameworks. This review serves as a valuable entry point for researchers and practitioners interested in advancing compound flood modeling and provides a solid foundation for supporting the development of more effective compound flood prediction tools. Please see the "specific comments" below for strengthening the manuscript.

**Reply:** We appreciate the reviewer's positive evaluation of the manuscript and the specific suggestions for strengthening it. We have revised the manuscript accordingly, addressing the comments through clarifications of the hybrid-model definitions, improvements to the descriptions of statistical and ML components, and refinements to several sections to improve coherence. Detailed responses to each comment are provided below.

**Reviewer Comment 2.2 —** As a key component of hybrid frameworks, the statistical modeling aspect appears to be less discussed in the manuscript. While they are included in Section 2, there is very little reference to them in sections 3.1 and 3.2 (and in section 4). In Figures 3 and 4 it's not clear where the statistical/data-driven component sits that makes it a hybrid model. It becomes a bit clearer further down and in the appendix, but it would help to strengthen the focus on the "hybrid" aspect since that is what the paper is about, otherwise some parts become more like a duplicate of the Santiago-Collazo et al. (2019) paper, explaining how different process-based models can be linked/coupled.

**Reply:** In the revised manuscript, we have strengthened the discussion of the statistical and machine-learning components within the hybrid modeling framework and clarified their roles throughout Sections 3.1, 3.2, and 4. Specifically:

- **Figures 3 and 4 (Sections 3.1 and 3.2):** Both captions and accompanying text have been revised to explicitly identify where the statistical or data-driven components sit within the sequential and feedback hybrid frameworks. For Figure 3, we clarify that the riverine component is driven by a statistical or ML-based hydrologic generator (e.g., stochastic rainfall, probabilistic runoff). For Figure 4, we now highlight how statistical/ML models may supply atmospheric, hydrologic, or oceanic boundary conditions in place of (or in combination with) full physics-based components.

**Hydrology and Earth System Sciences**
Notes on revision made to manuscript egusphere-2025-4623

- **Introduction:** We added text emphasizing the broader benefits of incorporating statistical/ML models within hybrid frameworks, beyond multivariate dependence modeling (as also noted in our response to Comment 2.7).
- **Section 3.1:** We included a discussion explaining how the boundary-condition requirements of different hydrodynamic model types (bathtub vs. dynamic) shape the role of statistical models, following your Comment 2.3.
- **Section 4.2:** We introduced an entirely new subsection dedicated to explaining the functional roles of statistical and ML components within hybrid models, including event generation, emulation/surrogacy, uncertainty propagation, and probabilistic hazard synthesis (see response to Comment 1.3).

Collectively, these revisions enrich the manuscript's focus on the hybrid (statistical–physical) nature of the modeling frameworks and clarify how statistical/ML components are embedded within hybrid approaches.

**Reviewer Comment 2.3 —** Related to the previous comment, for the sequential modeling approach, the role and implementation of statistical models may vary depending on the boundary condition requirements of the flood models (e.g., bathtub vs. dynamic). These statistical frameworks can range from generating only the peak boundary conditions to producing a full time series of boundary inputs (e.g., Moftakhari et al., 2019; Maduwantha et al., 2025 ). I encourage the authors to briefly discuss how such approaches can strengthen hybrid modeling frameworks.

**Reply:** We have revised the manuscript in Section 3.1 to cover these aspects:

"*An important consideration in sequential hybrids is that the design of the statistical block depends on the type and complexity of boundary conditions required by the downstream flood model. For bathtub-type (steady-state) inundation models, statistical frameworks typically generate only peak boundary conditions, such as maximum surge, river discharge, or storm rainfall, based on joint-probability analyses (Moftakhari et al., 2019). In contrast, dynamic hydrodynamic models require complete time-series boundary forcings (evolving surge levels, hydrographs, rainfall-intensity curves), which can be produced with stochastic event generators or probabilistic resampling approaches (Maduwantha et al., 2025). These flexible statistical-to-physics interfaces strengthen hybrid modeling by allowing the probabilistic backbone to support a spectrum of hydrodynamic*

> *solvers, ranging from highly simplified static methods to fully dynamic, process-based simulations. They also expand the capability of hybrid frameworks to explore a wider range of plausible compound events under both present and future climate conditions.*"

**Reviewer Comment 2.4 —** The section on ensemble hybrid modeling for compound flooding is conceptually sound and addresses a timely topic. The only example provided relates to the use of ensemble methods for rainfall forecasting, which, while relevant, does not fully reflect the complexity involved in applying such an approach to compound flood processes. There are likely to be practical challenges when implementing such ensemble approaches for compound flooding (e.g., can statistical modeling alone generate compound flood depths that are comparable to physics-based model outputs to get the final weighted prediction?). A further discussion on how such approaches can be practically implemented would strengthen the manuscript.

**Reply:** We agree with the reviewer that applying ensemble hybrid modeling to *compound flooding* introduces additional challenges that are not present in rainfall forecasting. The reason our manuscript only referenced a rainfall-forecasting example is that, to the best of our knowledge, there is currently no published study that has implemented an ensemble hybrid framework for compound flood depths. This gap is not due to a lack of interest but rather to the fact that existing hydrodynamic solvers are still mostly too computationally expensive to generate the large number of realizations needed for ensemble weighting, and because purely statistical models have not yet matched the spatial fidelity required for inundation.

However, recent developments indicate that ensemble hybrid modeling for compound flooding is becoming feasible, and we now discuss these pathways more explicitly in the revised text:

1. **Feedback hybrid models running in parallel with a process-based solver:** Recent advancements in PINN-based hydrodynamic models (e.g., Radfar et al., 2025) demonstrate runtimes that are *orders of magnitude* faster than even reduced-complexity solvers like SFINCS, while maintaining comparable accuracy. This enables running a PINN model alongside a traditional process-based model. In such a setup, the ensemble weights can be assigned based on metrics such as local error, computational time, and stability under rapidly varying conditions.

2. **Neural-network–based SWE solvers as independent ensemble members:** Neural-network SWE solvers (Chen et al., 2025) show that data-driven models can approximate the shallow water equations directly. These models bypass some of the stability constraints of numerical solvers and

produce physically consistent flood depths at very low computational cost. They can serve as additional fast ensemble members whose outputs are combined with the core hydrodynamic model.

3. **Application in digital-twin and multi-model environments:** With the computational improvements from both PINNs and neural SWE solvers, it becomes practical to maintain a suite of physics-based and ML-based models within a digital-twin framework. Ensemble weighting can then be based on performance, uncertainty, or run-time constraints.

We added this explanation to clarify that ensemble hybrid modeling for compound flooding is not yet demonstrated in the literature, but the emerging tools described above make it a realistic and timely research direction. Updated text in Section 3.3:

"*Although ensemble hybrid modeling has been explored in some hydrologic contexts, its application to compound flooding remains largely undeveloped. This gap is mainly due to the high computational cost of process-based compound flood models and the absence of fast, physically consistent surrogates capable of producing independent depth predictions at scale. Recent progress, however, is reducing these barriers. Feedback hybrid models, such as PINN-based flood solvers (Radfar et al., 2025), achieve runtimes orders of magnitude faster than reduced-complexity solvers, such as SFINCS, while maintaining comparable accuracy, making it feasible to run a PINN model alongside a traditional hydrodynamic model within an ensemble framework. Neural-network–based shallow-water equation (SWE) solvers (Chen et al., 2025) offer another emerging option that directly approximates the SWE at low computational cost and yields stable flood predictions. These developments point toward practical ensemble hybrid configurations in which fast physics-informed or neural-network solvers operate in parallel with process-based models, enabling performance-dependent weighting and allowing ensemble systems to capture the full range of compound flood dynamics. Recent updates in HEC-RAS (alpha version) allow practitioners to reduce numerical complexity in a widely used modeling platform, thereby expanding the set of tools available for operational flood modeling. These emerging pathways also integrate well with digital-twin environments, in which a suite of physics-based and ML-based models can operate within an ensemble modeling setup to provide (near)real-time compound flood forecasting and decision support.*"

**Reviewer Comment 2.5 —** Feedback hybrid models are recognized as models that enable bidirectional information exchange between coupled models/components. However, it is not clear whether bidirectional exchange between only two domains, such as atmospheric and ocean

models, is sufficient to categorize them as feedback-hybrid compound flood models. In regions where tides, storm surge, and river discharge interact, tightly coupled models (e.g., wave and ocean circulation models) can provide a more robust way to simulate flooding. However, would similar advancements be achieved when bidirectional information exchange exists only between atmospheric–ocean or atmospheric–land surface models? Same for the example given in line 251, soil moisture content is mentioned as influencing local weather conditions. While this is true, is that alone sufficient to classify the entire modeling approach as feedback?

**Reply:** In our framework, a modeling system is classified as a *feedback hybrid* when there is bidirectional exchange of hydrodynamically relevant information between process-specific sub-models, as defined in Section 2. These process-specific domains correspond to the primary drivers of compound flooding—atmospheric forcing, runoff and river discharge, coastal water levels, storm surge, and wave dynamics—and not to every possible variable or surface process that may influence the hydrologic cycle.

Therefore, the criterion for classifying a model as a feedback hybrid is not whether the feedback occurs between *all* domains involved in compound flooding, nor whether the feedback occurs in a specific "preferred" pair (e.g., wave–circulation vs. atmosphere–ocean). Instead, any bidirectional exchange between two or more process-specific components is sufficient to meet the definition, even if the coupling involves only a subset of the full system (e.g., atmosphere–ocean, ocean–river, river–land surface).

Whether that particular coupling configuration results in *improved* prediction of compound flooding is a separate question and depends on the regional context, the dominant flood drivers, and the objectives of the modeling study. Our classification deliberately separates:

(1) What qualifies as feedback coupling, versus

(2) Whether that feedback improves the representation of compound flood dynamics (Section 4.4, Step 6, Validation strategy).

To avoid confusion, we have added a clarifying explanation in Section 3.2:

"*In this framework, a model is categorized as a feedback hybrid when there is bidirectional information exchange between any two process-specific domains, consistent with the definition in Section 2. These domains correspond to the primary compound flood drivers—atmospheric forcing, watershed runoff, river discharge, coastal water levels, tides, storm surges, and waves. The required condition is therefore the presence of two-way updates between at least two of these driver-specific components, regardless of which pair is coupled. The extent to which a specific feedback*

*configuration improves compound flood prediction or process understanding depends on the regional context and the dominant flood drivers and must be assessed through a targeted validation strategy (see Section 4.4, Step 6), which then guides how future modeling setups should be refined or expanded*."

And also, Section 4.4, Step 6:

"6. Validation strategy: *Implement comprehensive validation protocols for both individual components and the integrated system, using observational data where available, and apply sensitivity testing to evaluate how model behavior responds to changes in the dominant flood drivers. Because the benefits of a particular configuration depend on the regional context and the relative importance of rainfall, river discharge, tide, surge, and wave processes, validation should explicitly assess whether the chosen coupling improves predictive skill or process representation under the conditions most relevant to the study domain*."

**Reviewer Comment 2.6 —** In Figure 3, it seems that it's just using discharge from a river model as input for a coastal model. According to their own definition, it would only be a hybrid model when one of the models is statistical/data-driven. That is not mentioned in the figure or caption. Same for other figures. I assume they mean that one of the models always has to be a statistical model but it's not clear. Or is it enough if one of the shown model components is itself linked to a statistical model, like a rainfall generator driving the hydrologic model, which then connects to the coastal model!?

**Reply:** We have updated the text and figure caption for clarity. Updated text:

"*Sequential hybrid models can be illustrated through compound flood modeling studies where river discharge and storm surge interactions are simulated (Figure 3). In a typical setup, a statistical or data-driven riverine model first generates the hydrologic inputs that drive downstream dynamics. This component may include, for example, a stochastic rainfall generator or a probabilistic runoff model that produces river discharge based on upstream flow measurements, infiltration capacity, and local rainfall data …*"

Updated caption:

"*Figure 3. Schematic illustration of a sequential hybrid modeling framework for simulating compound flooding in coastal river systems. The statistical or data-driven riverine model (left)*

> *generates hydrologic inputs (e.g., discharge derived from stochastic rainfall generation or probabilistic runoff estimation), which are then routed into a hydrodynamic model.*"

**Reviewer Comment 2.7 —** In line 365, the authors mention that the strength of the statistical component is its ability to be faster. However, in the introduction (line 86), the authors only mention the ability of statistical methods to model multivariate dependence as their main strength, without noting the advantage of being faster compared to physics-based modeling. I suggest adding this point to the introduction for consistency.

**Reply:** We have expanded the Introduction to clearly highlight this additional advantage of statistical/ML approaches:

> "*In addition, statistical or machine learning (ML) approaches can generate flood hazard predictions much faster than full physics-based hydrodynamic models, enabling rapid scenario evaluation and real-time applications (Anderson et al., 2021; Bass and Bedient, 2018; Radfar et al., 2025). Modern ML surrogates also offer improved generalizability and transferability, as some models can be applied across different regions or forcing conditions with minimal retraining, overcoming the site-specific configuration demands of traditional process-based models (Daramola et al., 2025a; Daramola et al., 2025b).*"

***References:***

Anderson, D. L., Ruggiero, P., Mendez, F. J., Barnard, P. L., Erikson, L. H., O'Neill, A. C., ... & Marra, J. (2021). Projecting climate dependent coastal flood risk with a hybrid statistical dynamical model. *Earth's Future*, *9*(12), e2021EF002285.

Bass, B., & Bedient, P. (2018). Surrogate modeling of joint flood risk across coastal watersheds. *Journal of Hydrology*, *558*, 159-173.

Daramola, S., Muñoz, D. F., Moftakhari, H., & Moradkhani, H. (2025). A cluster-based temporal attention approach for predicting cyclone-induced compound flood dynamics. *Environmental Modelling & Software*, 106499.

Daramola, S., Muñoz, D. F., Sakib, M. S., Thurman, H., & Allen, G. (2025). A transferable deep learning framework to propagate extreme water levels from sparse tide-gauges across spatial domains. *Expert Systems with Applications*, 130222.

Radfar, S., Maghsoodifar, F., Moftakhari, H., & Moradkhani, H. (2025). Integrating Newton's Laws with deep learning for enhanced physics-informed compound flood modelling. *arXiv preprint arXiv:2507.15021*.

**Reviewer Comment 2.8 —** In Figure 6, "structural flexibility" is listed as a key consideration under ensemble modeling. However, it is not clear what is meant by structural flexibility in this context, and how it specifically relates only to ensemble modeling over other approaches.

**Reply:** To avoid ambiguity, we have clarified its meaning in the text after Figure 6 by explicitly defining structural flexibility for ensemble models:

> "*Here, structural flexibility means that individual model predictions can be re-weighted (i.e., emphasized, down-weighted, or assigned zero weight), and that additional models can be added or existing ones replaced without altering any coupling interfaces, because these decisions occur at the output-combination stage.*"

We use structural flexibility only for ensemble models because this feature is specific to output-level coupling. In sequential and feedback hybrids, model components exchange boundary conditions or state variables during runtime through fixed data-exchange interfaces. Therefore, replacing a component in those modeling frameworks typically requires reconfiguring or recalibrating these couplers. In contrast, ensemble models combine results only at the output stage, allowing model predictions to be reweighted or replaced without modifying any internal couplers. That is why the term structural flexibility is used only for ensemble models.

**Reviewer Comment 2.9 —** Some of the relevant recent studies on compound flood modeling are missing from the review (e.g., Jane et al, 2022; Orton et al., 2020). Including them would improve completeness and strengthen the review's contribution.

**Reply:** As per your suggestion, we cited these papers in the revised manuscript as follows:

- **Introduction:** As such, univariate methods are inadequate for capturing interactions between multiple flooding drivers and do not consider the statistical dependence (i.e., joint probability) between multiple flood drivers (Jane et al., 2020).

**Hydrology and Earth System Sciences**

Notes on revision made to manuscript egusphere-2025-4623

- **Section 3.1:** Sequential process-based implementations also include studies where rainfall-driven tributary inflows are routed into coastal hydrodynamic storm-tide models. For example, (Orton et al., 2020) combined watershed-derived runoff with a 3-D storm-tide simulation in the Hudson estuary to assess tidal-river flood hazards.